# What Are Tools Anyway?
# A Survey from the Language Model Perspective

**Zora Zhiruo Wang**[♦]     **Zhoujun Cheng**[♦]     **Hao Zhu**[♦]     **Daniel Fried**[♦]     **Graham Neubig**[♦]
[♦]Carnegie Mellon University     [♦]Shanghai Jiao Tong University
{zhiruow,dfried,gneubig}@cs.cmu.edu

## Abstract

Language models (LMs) are powerful yet mostly for text generation tasks. Tools have substantially enhanced their performance for tasks that require complex skills. However, many works adopt the term "tool" in different ways, raising the question: *What is a tool anyway?* Subsequently, *where and how do tools help LMs?* In this survey, we provide a unified definition of tools as external programs used by LMs, and perform a systematic review of LM tooling scenarios and approaches. Grounded on this review, we empirically study the efficiency of various tooling methods by measuring their required compute and performance gains on various benchmarks, and highlight some challenges and potential future research in the field.[1]

## 1 Introduction

Language Models (LMs) have become increasingly effective in solving text-generation tasks, by taking in natural language (NL) instructions from users and outputting NL responses, such as answering the "What is the capital of the US?" with "Washington D.C.". However, LMs often struggle to perform tasks that require complex skills (e.g., math or complex reasoning), and are fundamentally unable to solve other tasks that require access to information not in their training data or parametric knowledge (e.g., the current weather or date).

To solve this problem, researchers and practitioners are turning to LMs enhanced with *tools*, which help *facilitate* the task-solving process of LMs, or *extend* LMs with new abilities that the LM does not possess otherwise (Qin et al., 2023; Mialon et al., 2023). For example, a `calculator` tool can facilitate math calculations, or a `get_time()` tool can enable LMs to obtain time. Inspired by the tools used by humans (Shumaker et al., 2011), some works introduce application-specific `software` as tools, such as a `search engine` to obtain knowledge (Lazaridou et al., 2022; Komeili et al., 2022) or a `translator` to process unknown languages (Schick et al., 2023).

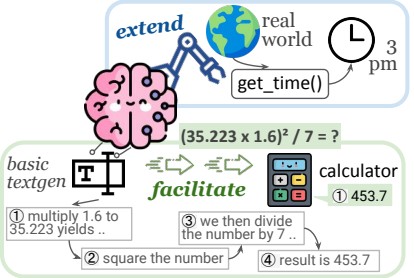

Figure 1: Illustration of tools extending and facilitating LM task-solving.

With the development of numerous application programming interfaces (APIs) on the web, many works collect `APIs` as tools to access world data in real-time (Balog et al., 2016; Xu et al., 2023; Qin et al., 2024) and via multiple modalities (Tang et al., 2023). Instead of black-box APIs, other works use crafted `functions` to query over structured tables (Wang et al., 2024a; Cao et al., 2023) or images (Surís et al., 2023), where tools can be created by human (Gupta & Kembhavi, 2022) or model experts (Wang et al., 2023a; Cai et al., 2023; Wang et al., 2024b).

However, despite this broad and burgeoning area of tool use in LMs, existing surveys only cover certain tool categories such as software (Mialon et al., 2023) or APIs (Qin et al., 2023). In this paper, we provide a unified view of tool use across a broad range of scenarios.

---

[1]https://github.com/zorazrw/awesome-tool-llm

We start with proposing *a unified definition* of tools and explain *why tools help task-solving* (§2). We present the *basic tool-use paradigm* (§3) and study a variety of tool-using scenarios by enumerating *which tools exist* and *to which tasks they apply* (§4). Next, we study advanced approaches for *complex tool usage* and even *make new tools* if they are unavailable for the task (§5). We then summarize existing testbeds and evaluation metrics across LM tooling works, and highlight several missing aspects with concrete metric suggestions (§6). Lastly, grounding on our empirical analysis about *when tools are effective*, we identify the most efficient tooling approaches and the tasks that benefit most from tools (§7).

## 2 Background

### 2.1 What are tools?

Because LMs are products of the digital world, tools employed by LMs are often computer **programs** that are executable in corresponding environments. Referring back to animal-used tools Shumaker et al. (2011) defined as *"the external employment of an unattached or manipulable attached environmental object."* Similar to human-used physical tools, LM-used program tools should also be **external** to the employer (i.e., the LM) and are part of the environment. Also, instead of arbitrary program snippets, a tool is a **function** (e.g., `plus_one`), meaning that it can be applied to other objects (e.g., data) and yield an output (e.g. `plus_one(1) → 2`).

Existing definitions of LM-used tools touch on some of these aspects. Qin et al. (2023) appeal to the similarity to human tool use, but do not define what entails a tool. Mialon et al. (2023) define *a tool* as *"an external module that is typically called using a rule or a special token and whose output is included in the augmented LM's context."* We argue for a somewhat broader definition than this, which encompasses a wide variety of more recent works on tool usage:

**Definition 1.** *An LM-used tool is a function interface to a computer program that runs externally to the LM, where the LM generates the function calls and input arguments in order to use the tool.*

### 2.2 Why are tools helpful?

Tools can help task-solving in mainly three ways, as reflected by their functions: perception, action, and computation. A tool may belong to one or more of these three categories.

**Perception**    Perception tools provide or collect information from the environment. An example is using a `get_time()` API to obtain the current time, which is not included in the LM's parametric knowledge learned from training.

**Action**    Action tools can exert actions on the environment and change its state. For example, executing `make_post(website, post)` can change the content on a `website`.

**Computation**    Computation tools do not perceive or modify the external environment, but use programs to tackle complex computational tasks. For example, a `calculator` is a computation tool for mathematical calculation. Note that the computation also includes more general acts of computing beyond numerical calculation. Therefore, a `translator` is also a computation tool that can be used to translate between languages.

A tool can fall into multiple categories. For instance, a `search` tool can perform both computation and perception: it computes document similarity to find relevant ones, but also perceives the environment (the web) and fetches data (returned documents) from it. In a similar spirit, SQL queries can be used as computation tools (e.g., `SELECT SQRT(16) / 10 AS result`), perception tools for viewing data (e.g., `SELECT name FROM data`), action tools to modify data (e.g., `INSERT INTO data VALUES name`), or all of the above (e.g., `INSERT INTO counts (grp_id, grp_cnt) SELECT grp_id, COUNT(*) FROM data GROUP BY grp_id`).

### 2.3 Tools and "Agents"

There has recently been a burgeoning of work on LM-powered agents (Xi et al., 2023; Sumers et al., 2024). Russell & Norvig (2010) define agents as *"anything that can be viewed as perceiving its environment through sensors and acting upon that environment through actuators."* According to this definition, agents are programs that use perception tools to perceive the

situated environment, or action tools to interact with the environment. Models that only use computation tools and do not interact with their environments through perception or action tools arguably do not fall under the category of "agents" according to this definition.

# 3 The basic tool use paradigm

We show an illustrative example of a basic tool-use paradigm introduced by Schick et al. (2023), which many tool-related works adopt (Figure 2).

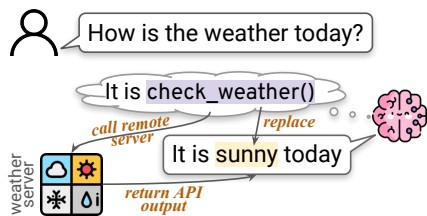

Assuming an LM communicates with users mainly in NL, upon receiving a user query such as "How is the weather today?", the LM then proceeds to generate either text or tool calls. In the example, the LM starts with generating a few tokens of text "It is ...". When the LM needs to seek external tools to complete the task, e.g., get real-time weather information, it generates tokens of the tool name and corresponding input arguments check_weather() to construct a complete tool calling expression. This expression will trigger a shift from text-generation mode to tool-execution mode. The tool-hosting server executes the expression and

Figure 2: The basic tool use paradigm. LM calls `check_weather` tool by generating text tokens. This call triggers the server to execute the call and return the output `sunny`, using which the LM replaces the API call tokens in the response to the user.

returns the execution output "sunny" to the LM. The returned output replaces the tool call in the LM-generated tokens (from "It is check_weather()" to "It is sunny"). Accordingly, LM shifts back to the text generation mode and continues to finish the response by generating new text tokens, e.g., adding "today." Finally, the LM returns the response to the user.

In order for LMs to use this basic paradigm of using tools, current works mainly leverage inference-time prompting and training-time learning methods.

**Inference-time prompting** Leveraging the ability of LMs to learn in-context (Brown et al., 2020), many works provide tool information through a prompt and expect LMs to acquire abilities to use these tools from input contexts. This is achieved by providing instructions about the task, example pairs of queries and solutions that use tools (Gupta & Kembhavi, 2022; Lu et al., 2023a; Paranjape et al., 2023; Shen et al., 2023a; Yang et al., 2023), and/or documentation of the tools' functionality (Hsieh et al., 2023).

**Learning by training** Beyond learning tools from test-time contexts, LMs can learn from examples that use these tools during training. LMs can simply be trained to generate tool-using solutions, where the examples can be manually annotated by humans (Li et al., 2023), synthesized by larger teacher LMs (Tang et al., 2023; Qin et al., 2024; Huang et al., 2024), or bootstrapped by the test-time LM itself (Schick et al., 2023).

# 4 Scenarios where tools are useful

While LMs may easily solve many tasks without tools, many other tasks greatly benefit from tool use. In this section, we study a broad range of scenarios where tools have been used to assist agents. We discuss tasks where human-created, task-specific tools can improve their performance (§4.1), as well as scenarios where tools may not be as useful (§4.2).

## 4.1 Utilizing existing tools for specific applications

While it is difficult to exhaustively enumerate every scenario where tools could be useful, we summarize some major categories of tool use in Table 1 and below.

📖 **Knowledge access** LMs store limited knowledge during training due to both limits in (i) the data that they are trained on and (ii) the ability of LMs to accurately memorize and utilize their training data. Knowledge-accessing tools can help alleviate this issue. SQL and SPARL executors can provide access to data in structured knowledge bases (Thoppilan et al., 2022; Hao et al., 2023) or graphs (Zhuang et al., 2023). An search engine over the

| Category | Example Tools |
|---|---|
| 📖 Knowledge access | `sql_executor(query: str) -> answer: any`
`search_engine(query: str) -> document: str` |
| 🏛 Computation activities | `calculator(formula: str) -> value: int | float`
`python_interpreter(program: str) -> result: any`
`worksheet.insert_row(row: list, index: int) -> None` |
| 🌐 Interaction w/ the world | `get_weather(city_name: str) -> weather: str`
`get_location(ip: str) -> location: str`
`calendar.fetch_events(date: str) -> events: list` |
| 🎞 Non-textual modalities | `cat_image.delete(image_id: str) -> None`
`spotify.play_music(name: str) -> None`
`visual_qa(query: str, image: Image) -> answer: str` |
| ⚙ Special-skilled LMs | `QA(question: str) -> answer: str`
`translation(text: str, language: str) -> text: str` |

Table 1: Exemplar tools for each scenario. A tool may fall into one or more categories.

web (Yao et al., 2023; Schick et al., 2023; Paranjape et al., 2023) can enable LMs to access more up-to-date information (Komeili et al., 2022; Lazaridou et al., 2022). More generally, retrieval-augmented systems (Asai et al., 2023) can be seen as using a `retriever` tool.

🏛 **Computation activities**  Complex computing activities such as math calculations are known to be challenging for neural LMs (Schick et al., 2023). While even a `calculator` can enhance LMs' numeracy abilities (Parisi et al., 2022; Hao et al., 2023), more generic `Python` programs are also employed to aid reasoning tasks (Gao et al., 2023b; Chen et al., 2023; Wang et al., 2023b). For more complex professional jobs, business tools are also applied, such as using `worksheet` to manipulate Google Sheets (Xu et al., 2023), or even tools for financial, medical, education, or advertising domains (Tang et al., 2023; Huang et al., 2024).

🌐 **Interaction with the world**  LMs without tools are fundamentally unable to perceive and act in the world around them, necessitating tools that can interact with the world. For instance, tools can help LMs access real-time information such as weather (Xu et al., 2023; Tang et al., 2023), or positional knowledge such as location (Qin et al., 2024). Meanwhile, tools can enable LMs to take actions on the world such as managing calendars (Schick et al., 2023) and emails (Qin et al., 2024). Beyond web-based activities, tool-augmented LMs can engage in physical activities in embodied environments, such as fishing with rods or mining with axes in the Minecraft world (Wang et al., 2023a); further propagate to the real-world tasks to perform cooking (Singh et al., 2022; Shridhar et al., 2020), plotting (Liang et al., 2023), and even conducting chemical research (Boiko et al., 2023).

🎞 **Non-textual modalities**  While many LMs only consume and generate texts, some works bring in access to visual (Gupta & Kembhavi, 2022; Surís et al., 2023), audio (Yang et al., 2023; Gao et al., 2023a), or other modalities. For example, LMs can access images with `cat_image` APIs (Xu et al., 2023; Tang et al., 2023) or songs (Huang et al., 2024) provided by `spotify`, even answer questions about them (Gupta & Kembhavi, 2022; Gao et al., 2023a).

⚙ **Special-skilled LMs**  Some works propose to use specialized LMs as tools, essentially using the main LM as a task planner to dispatch requests to other LMs. Schick et al. (2023) propose QA models to fill in factoid details in responses, Thoppilan et al. (2022); Schick et al. (2023); Paranjape et al. (2023) use machine translation models to assist multilingual tasks. Beyond specific tasks, some works adopt multiple neural models from Hugginface or similar platforms (Patil et al., 2023; Shen et al., 2023a), or further fine-tune them on various data (Viswanathan et al., 2023). Compared to the base LM, these tool models mainly vary in their specialized skills, and may or may not have architectural differences to the base LM.

## 4.2 Where are tools *not* useful?

Despite that tools can be helpful under many scenarios discussed above, it is also important to note scenarios where tools are arguably not very helpful. Some example tasks where tools have not (yet) been used to great effect include machine translation, summarization,

and sentiment analysis (among others). These are tasks that are not easy to perform using non-ML methods (c.f. accessing databases can be done using SQL), and can be performed with high accuracy by a powerful LM alone. One intuitive reason is that the tools currently leveraged for these tasks are *neural networks* and have limited advantages over the base LM.

## 5    Advanced tool-use methods

Given this understanding of the basic tooling paradigm and the scenarios in which tools are useful, we now discuss more advanced approaches for tools. Concretely, we study multi-tool selection and complex usage (§5.1), tooling under programmatic contexts (§5.2), and creation of tools when they are not available a-priori (§5.3).

### 5.1    Complex tool selection and usage

Depending on the number of tools available, the system may include an implicit or explicit tool selection module. If tools are already *designated* for the task (Lazaridou et al., 2022; Thoppilan et al., 2022), then no tool selection is needed. If *a small number* (e.g., 5–10) of tools are available, metadata and use cases of these tools can be provided as input contexts along with the user query (Schick et al., 2023; Paranjape et al., 2023), and LMs can directly select tools from contexts via a standard generation process. If the toolbox size *further grows* (e.g., to hundreds), fitting all tools into model inputs is not feasible. Thus an extra retrieval step is often incorporated: a retriever model short-lists the most relevant tools and feeds their metadata to the solution-generation LM. Specifically, Zhou et al. (2023); Qin et al. (2024) train retriever models that map NL to tool documentation. Yuan et al. (2023) ask LMs to write hypothetical descriptions then find similar tools. More easily, some directly use off-the-shelf embeddings (Meng et al., 2024; OpenAI) or training-free retrievers (Robertson et al., 2009).

For complex queries that require multiple tools to solve, the common approach so far is to break down the task and sequentially tackle each step by using a single tool (Paranjape et al., 2023). However, this sequential multi-step paradigm may not be reflective of more complex or realistic tool usage. For example, a user may prefer *nested* function calls `check_weather(get_local_time(‘Pittsburgh’))` to allow information hiding or encapsulation (Rogers, 2001), *parallel* calls to reduce round trips with the API (Eleti et al., 2023), or *iterative* calls of `buy_ticket(event)` until it returns `True` to indicate a successful transaction.

### 5.2    Tools in programmatic contexts

Unlike text-based tasks where tools are auxiliary modules to LMs, on programmatic tasks, where code LMs can solve the problem by generating programs, tools can be seen as compositions of basic functions. In this part, we discuss tools in programmatic tasks for domain-specific (§5.2.1) and general-purpose problems (§5.2.2).

**Focus on varied tools**    Depending on the tasks of interest, existing works focus on different types of tools under programmatic contexts. With the increasing complexity of these tools and presumably a decreasing familiarity of LMs about them, there are works that adopt (i) *built-in functions* of a programming language (PL) to augment LMs in symbolic reasoning, (ii) *external libraries* in pre-designed packages to tackle complex open-domain coding queries (Wang et al., 2023c), and (iii) *utility functions* unseen at training time to solve specific tasks.

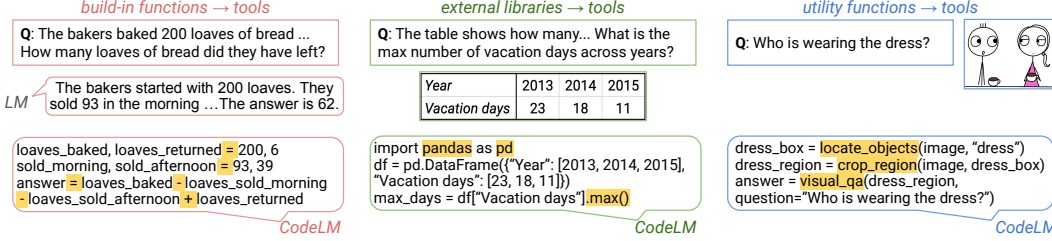

Figure 3: Tools involve built-in functions, external libraries, or task-specific utility functions.

### 5.2.1 Domain-specific semantic parsing

NL-to-code generation systems have been studied for many years on special-domain tasks such as querying databases (Zelle & Mooney, 1996; Zettlemoyer & Collins, 2012) or knowledge graphs (Berant et al., 2013). Code produced by these systems is often domain-specific logical forms (DSL) designed by experts, such as lambda expressions (Liang, 2013), SQL queries (Yu et al., 2018), or the recent QDMR grammar (Wolfson et al., 2020). Many agentic tasks also adopt DSL to operate in task-specific environments, such as `click` or `type` in web navigation (Liu et al., 2018; Yao et al., 2022; Zhou et al., 2024), `placeItem` in the embodied Minecraft world (Wang et al., 2023a), or `set_joint_target` for robots (Yu et al., 2023). While DSL *built-in actions* can be directly used, for complex queries, solution programs with basic DSLs alone can be hard to interpret or cumbersome to use.

### 5.2.2 General-purpose code generation

Recent code generation systems have expanded from using DSL to more general-purpose PLs such as Python or Java (Yin & Neubig, 2017; Chen et al., 2021). These languages enable more programming flexibility and readily apply to versatile scenarios. As we have introduced using *built-in actions* as tools in §4.1, we discuss more on two other common categories of tools for code LMs, namely *external libraries* and task-specific *utility functions*.

**External libraries** From the usage of PLs, built-in functions are internal to whichever environment, whereas third-party libraries lie externally and need to be imported to tackle specific contexts, such as Figure 3 (middle). Aligning with this conception, Zhang et al. (2023) use Python libraries such as `matplotlib` to plot figures and `pandas` to manage data.

**Utility functions** Expert-crafted utility functions, usually unseen at training time, can also be used as tools. E.g., in Figure 3 (right), the highlighted `locate_objects` (Gupta & Kembhavi, 2022) loads neural models to detect object regions in images. Also, Cheng et al. (2023) use GPT as a tool to query world facts external to table contents, Cao et al. (2023) further design macro APIs for advanced tabular operations. However, because human tool creation requires expertise and effort, some works explore using LMs to create new tools.

### 5.3 Tool creation and reuse

While one can readily use tools for tasks equipped with pre-designed tools, for tasks that do not have readily-applicable, human-created tools, some works explore using LMs to make tools and use them.

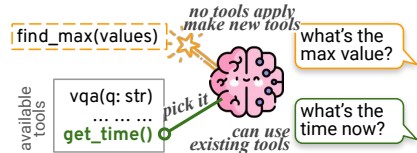

Figure 4: LM makes tools when no tools readily apply to the task.

**Domain-specific library abstraction** Works using DSLs often compose frequently-used-together actions as shortcut tools. Ellis et al. (2023) learn function abstractions such as `length` from lambda primitives (e.g., 0, +) for list or string tasks. Wong et al. (2021); Bowers et al. (2023) build libraries bottom-up from a corpus of DSL programs. Grand et al. (2023) use LLMs to abstract libraries with auto-documentation. For agentic tasks, Liu et al. (2018) learn common workflows to guide web navigation, e.g., composing basic {`click`, `like`} actions to form a higher-level login action `click(like('login'))`.

**General-purpose tool making** On general-purpose PLs, the DSL-oriented methods above may explode the search space and have limited utility. Instead, recent works leverage LMs' procedural knowledge to guide search. To start, Wang et al. (2023a) design an automatic curriculum in Minecraft to make and use Java program tools. LATM (Cai et al., 2023) use LMs to build, verify, and use Python tools on BigBench (bench authors, 2023). CREATOR (Qian et al., 2023) extend tool-making to math and table tasks, and improves task success by creating tools yet repetitively for every example, thus CRAFT (Yuan et al., 2023) add heuristic-based training to reduce tool repetition. Towards more efficient pipelines, ReGAL (Stengel-Eskin et al., 2024) learns from refactoring existing programs, while TroVE (Wang et al., 2024b) utilize inference-time execution signal and induces reusable tools on-the-fly.

# 6 How to evaluate tool use?

In this section, we study existing LM-tooling benchmarks (§6.1) and their evaluation metrics (§6.2), lastly, we discuss the missing yet important evaluation aspects of tools (§6.3).

## 6.1 Testbeds for evaluating tools

LM tool use can be evaluated on (i) repurposed existing datasets that can additionally benefit from tools (§6.1.1), and (ii) newly crafted benchmarks that necessitate tool use (§6.1.2).

### 6.1.1 Repurposed Existing Datasets

Many tasks are solvable by using LMs, yet often with great difficulty or inefficiency. Therefore, some works use tool-augmented LMs as an alternative approach to solve these tasks.

Most of these datasets require **reasoning**. Starting from NL questions that demand complex reasoning with Big-bench (bench authors, 2023), mathematical problems with MATH (Hendrycks et al., 2021), to answering world facts in NaturalQuestions (Kwiatkowski et al., 2019) and TriviaQA (Joshi et al., 2017). Beyond free-form texts, datasets with **structured data** also benefit from tools, particularly table-based QA with tabular math world problems in TabMWP (Lu et al., 2023b), Wikipedia tables in WTQ (Pasupat & Liang, 2015), and complex-structured tables in HiTab (Cheng et al., 2022). Beyond text modality, datasets involve **other modalities** benefit from modality-extending tools, e.g., answering questions about an image in GQA (Hudson & Manning, 2019), or image pairs in NLVR2 (Suhr et al., 2019).

Because tool use is an alternative method to solve these datasets, evaluations of these tool-augmented systems follow the standard evaluation process of these datasets, i.e., answer exact match. Note that to obtain the answers, all tool calls **need to be executed**, since the execution outputs are integrated into the final answers, as introduced in §3.

### 6.1.2 Aggregated API Benchmarks

Existing benchmarks can only benefit from a limited set of tools, yet there are far more tools we can utilize to perform versatile tasks, particularly the web APIs created by human developers. In Table 2, we list recent benchmarks for using APIs from various sources.

| Benchmark | Tool Source | Example Curation | Domain (§4.1) | Executable |
|-----------|-------------|------------------|---------------|------------|
| ToolBench$_1$ | existing dataset | adopted, human annotated | 🏦, 🌐 | ✓ |
| ToolBench$_2$ | RapidAPI | model synthesized | 🏦, 🌐 | ✓ |
| ToolQA | existing dataset | model synthesized | 🏦, 🈯 | ✓ |
| ToolAlpaca | PublicAPIs | model synthesized | 🈯, 🏦, 🌐, ▦ | ✗ |
| API-Bank | PublicAPIs | human annotated | 🏦, 🌐 | ✓ |
| MetaTool | OpenAI Plugins | model synthesized | 🏦, 🌐, ▦ | ✗ |
| Gorilla | HF, Torch, TF | model synthesized | 🎛 | ✗ |
| HuggingGPT | HF | human annotated | 🎛 | ✗ |
| Task Bench | HF, PublicAPIs | model synthesized | 🎛, ▦, 🌐 | ✗ |

Table 2: Benchmarks that use API tools to solve tasks. HF is short for HuggingFace.

**Tool sources**    Tools are mainly aggregated from existing datasets or public APIs. Benchmarks aggregating existing datasets (Xu et al., 2023; Zhuang et al., 2023) are often limited in domains. Other works scrape APIs from online sources such as Public APIs (Tang et al., 2023), RESTful APIs (Tang et al., 2023), or the OpenAI plugin list (Huang et al., 2024). Beyond human-written APIs (Li et al., 2023), neural models from ML platforms can be similarly formatted as APIs (Patil et al., 2023; Shen et al., 2023a;b). However, as tools come from heterogeneous sources, it is hard to select the best from or unify all these benchmarks.

**Example curation**    While most examples adopted from existing datasets are human annotated (Xu et al., 2023), only Li et al. (2023) do so for scraped APIs, by surveying 500 people and creating 314 dialogues manually. Most other works prompt GPT models to

synthesize examples (Qin et al., 2024; Tang et al., 2023; Shen et al., 2023b; Zhuang et al., 2023; Huang et al., 2024), however, leading to issues of *naturalness* and *executability*.

**First**, LMs are used to create examples given a heuristically selected set of tools, leading to two issues: (i) users may not use the selected tools together in practice, and (ii) the synthesized examples may not reflect the *natural use cases* of these tools. **Second**, 5 out of 9 benchmarks in Table 2 do not support tool execution, often to alleviate the cost of hosting unstable APIs, which cause *issues in evaluation*. Instead of matching final responses using lexical- (Li et al., 2023) or neural-based metrics (Tang et al., 2023; Qin et al., 2024), works resort to pseudo matching of API calling expressions with lexical (Tang et al., 2023; Shen et al., 2023a; Huang et al., 2024) and syntactical (Patil et al., 2023; Shen et al., 2023b) means.

## 6.2 What metrics are measured now?

**Task completion** Tools are used to assist task solving. Most works that allow tool execution evaluate the task completion score to quantify the effectiveness of utilizing tools.

**Tool selection** Another common metric is the accuracy of selecting the correct tools. More concretely, it can serve as a measure of LM planning abilities — the process of breaking down a task into multiple steps and selecting tools to complete individual steps.

**Tool reusability** While tool reusability is often deemed important in took-making approaches (Cai et al., 2023; Yuan et al., 2023), only Wang et al. (2024b) evaluates it by the size of toolboxes. Reusable tools have generic functions, can be efficiently (re)used multiple times, and facilitate human verification in speed and accuracy aspects (Wang et al., 2024b).

## 6.3 What properties are missing?

**Efficiency of tool integration** The benefits of tools come with the cost of extra computation, particularly for teaching LMs to use tools via training or prompting. Besides performance gain, reporting the computation overhead can enable fairer comparisons of different works.

**Quality of tools** Beyond task accuracy, the *performance of tools* themselves is also important. Tool performance covers multiple aspects such as returning results quickly, using less compute, and not failing unexpectedly. One way to measure these aspects is to conduct API testing (Yasar, 2022; Ehsan et al., 2022) on their runtime, memory usage, and success rate.

**Reliability of unstable tools** Particularly for tools that involve *neural models* or *randomized components*, their output quality may be unstable and unpredictable. For example, the VQA tool (Gupta & Kembhavi, 2022) may answer some questions correctly but others incorrectly. It is essential to *be aware of* and reduce this uncertainty in contrast to stable, rule-based tools.

**Reproducible testing** Many world-interacting tools may return different results at different times. E.g., check_weather may return "sunny" today but "cloudy" tomorrow. This irreproducible behavior poses great challenges to create *static evaluation* benchmarks with reference answers. While some works alleviate this by evaluating API calls without executing them, a more rigorous method could be *parallel testing* (Sharma et al., 2018) — executing model-generated and reference programs in parallel and matching their final outputs.

**Safe usage** Many systems may only opt to use tools if they are trusted to be secure (Barbir et al., 2007). At the very least, users favor tools that can be easily understood and verified. Yet there may be more security issues such as authentication and data integrity (Ehsan et al., 2022). We encourage readers to peruse the referenced works above for thorough studies.

## 7 Trade-offs in tool usage

Using tools often brings better performance, however, should we always use tools? More concretely, is the performance gain from using tools worthy of the computation cost spent for LMs to learn tools? In Table 3, we empirically study the performance gain and learning cost of various methods on their tested datasets, from which we discover more efficient (i.e., achieve greater gains with less compute) methods and tasks that benefit more from tools.

For each work and each dataset they experimented with,[2] we measure the performance gain after LM learned or made tools to solve tasks, against baseline LMs with no prior exposure to tool-related information. We also quantify the computation cost of these methods in token-consuming training and inference processes. More computation details in §A.

| Type | Method | Task | Δ Perf. | # Params (B) | # Tokens (M) train | test |
|------|--------|------|---------|--------------|---------|------|
| tool use | ToolFormer | cloze | + 14.7 | 6.7 | 642.1 | 269.0 |
| | | math | + 30.4 | 6.7 | 3864.2 | 421.0 |
| | | QA | + 5.8 | 6.7 | 1101.2 | 189.0 |
| | | multilingual | - 0.2 | 6.7 | 606.0 | 274.0 |
| | | temporal | + 13.0 | 6.7 | 508.8 | 202.0 |
| | API-Bank | API | + 24.4 | 7 | **190414.6** | 0.0 |
| | ToolAlpaca | API | + 45.2 | 7 | **241889.3** | 0.0 |
| | Chameleon | science | + 2.6 | - | 0.0 | 88.3 |
| | | table | + 1.9 | - | 0.0 | 325.9 |
| tool making | LATM | BigBench | + 29.1 | - | 28.5 | 4720.0 |
| | CREATOR | math | + 4.5 | - | 0.0 | 5113.6 |
| | | table | + 0.0 | - | 0.0 | **6827.6** |
| | CRAFT | math | + 13.2 | - | 4126.6 | 4098.5 |
| | | table | + 17.2 | - | 2750.6 | 5018.2 |
| | TroVE | math | + 21.0 | - | 0.0 | 1825.2 |
| | | table | + 12.0 | - | 0.0 | 1358.8 |

Table 3: Computation cost (number of tokens in *M* and parameters in *B*) of tooling methods and their performance gain on experimented datasets. To fairly compare costs on datasets with different sizes, we report the average number of tokens spent on a testing example.

**What tasks benefit the most from tools?** In general, tasks that cover multiple domains experience the highest increase, such as ToolAlpaca in tool-using and BigBench in tool-making scenarios. Substantial gains may be expected on API benchmarks, because all examples are synthesized use cases for designated tools (§4.1), no-tool baselines are deprived of necessary components (i.e., tools) to solve the task, thus achieving much lower accuracy.

On existing datasets, the ToolFormer method in Figure 5 is the most efficient on MATH problems, showing the biggest 30.4 increase with little computation. While other tasks improve less, multilingual tasks even degrade by −0.2 points, despite using a similar amount of compute. This variance across tasks aligns with our discussion in §4: tasks necessitating ML-based tools benefit less from them.

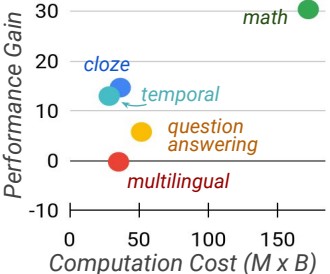

Figure 5: ToolFormer method on various tasks.

**What methods are efficient in tool-making?** While it is hard to fairly compare works testing on different datasets, the three tool-making methods (Creator, CRAFT, TROVE), experiment on the same MATH and TabMWP datasets in Figure 6 thus enable fair comparisons. TROVE is the most efficient method, costing only 1.2–1.4K tokens while increasing the accuracy by 12.0–21.0 points. In contrast, CREATOR and CRAFT are less efficient, costing 3.8–6.0 times of compute, yet achieve only minimal (0.0–4.5%) or comparable (4.1–5.0%) accuracy increases.

**Training-time vs inference-time cost** Training-time and inference-time costs may not be equally important to many practitioners, since inference may be run many times but training often only needs to be done once. If we only consider inference-time cost in Table 3,

---

[2]We did not measure some works due to insufficient resources.

the efficiency ranking of tool-using methods changes: Tool-Former requires more compute than API-Bank and ToolAlpaca. If the user has sufficient budgets for training and higher demand on inference-time efficiency, the training approaches proposed by API-Bank and ToolAlpaca could be more suitable.

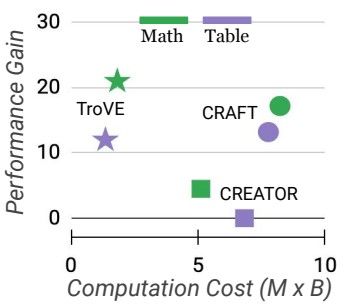

Figure 6: Comparing different tool-making methods.

## 8  Final Remarks

Our survey presents a definition for LM-used tools, various scenarios to apply tools, systematic summaries of existing methods, and an empirical analysis to guide when (on what tasks) and how (use what methods) one should use tools. We believe tools can greatly extend and facilitate LM abilities, and hope our work elicits more discussions and research developments in (i) curating benchmarks with natural use cases and executable tools, (ii) using multi-faceted evaluation metrics as in §6, and (iii) exploring more realistic and challenging scenarios for tool-using and tool-making techniques.

### Acknowledgments

We thank Saujas Vaduguru, Sherry Tongshuang Wu, Jiawei Liu, Shihao Liang, Pengfei Liu for the helpful discussions. Zora Zhiruo Wang is supported by the CMU Teng Family Presidential Fellowship. Hao Zhu is supported by NSF EAGER Award #2141751.

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

# A Detailed computation process for tooling trade-offs

For each method measured in §7, we describe the detailed processes in estimating their computation cost and performance improvement. For open-source models, we estimate cost $C = 6ND$, where $N$ is the number of tokens and $D$ is the parameter size (Figure 7, left). Because the parameter size $D$ of closed-source GPT is unknown, we only measure the number of extra tokens $N$ per example (Figure 7, right).

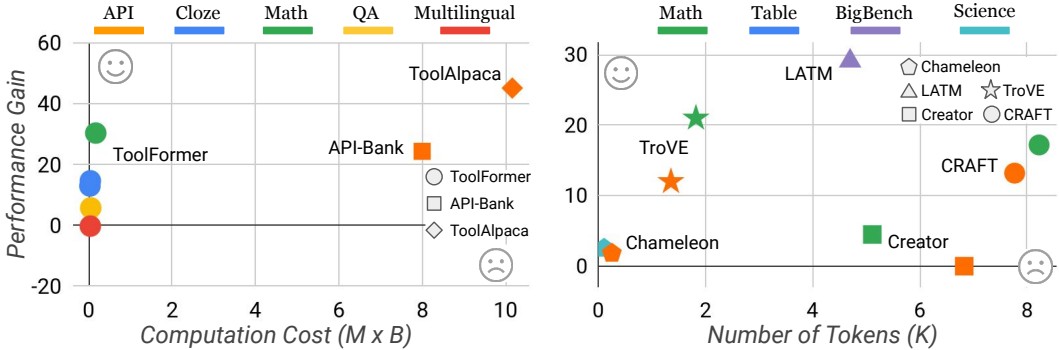

Figure 7: Computation cost of different approaches using open-source (left) and closed-source (right) models, and their performance gain on experimented datasets. We use different colors to represent tasks and different shapes to represent methods.

## A.1 Methods using known-sized models

For methods using models whose parameter sizes are known, we estimate the computation cost by the FLOPs during any additional modules such as training and inference with additional context. In general, the computation cost is majorly affected by (1) the number of tokens processed, and (2) the parameter size of models.

**API-Bank (Li et al., 2023)** This work trains the Lynx model that uses tools to solve problems in the proposed API-Bank dataset. The Lynx model is initialized by Alpaca 7B parameters, and trained on the API-Bank training set with 3 epochs. Therefore, we adopt the Alpaca 7B as the baseline and Lynx as the tool-using model, where the 3-epoch training is the additional computation cost introduced to enable tool use. We calculate the total number of tokens involved in the training process, including the example i/o and additional instructions. Because the baseline and proposed method use the same prompt at inference time, no additional computation is required. Regarding task performance, we adopt the total correctness across all evaluation systems, as reported in Table 3. We report the difference between the fine-tuned Lynx-7B and the zero-shot Alpaca-7B.

**ToolAlpaca (Tang et al., 2023)** This work proposes the ToolAlpaca dataset and trains Vicuna models to use tools. The baseline models are Vicuna-7B and Vicuna-13B models. The trained tool-using models are called ToolAlpaca-7B and ToolAlpaca-13B models. All ToolAlpaca models are trained on the training split for 3 epochs, so we estimate the cost during this training process for 7B and 13B models, respectively. We adopt the 'overall' results reported in Table 3, on examples with both simulated tools and real-world APIs, and report their average results. We measure the performance gain by the difference between the ToolAlpaca-7/13B and Vicuna-7/13B.

**Toolformer (Schick et al., 2023)** This work integrates five tools — question answering system, calculator, Wikipedia search, machine translation system, and calendar — respectively for five tasks transformed from a subset of CCNet (Wenzek et al., 2020). Starting with GPT-J models (Wang & Komatsuzaki, 2021) as the no-tool baseline, they train on $25k$ model-synthesized examples for each tool and obtain the Toolformer models, causing a total of $1M$ FLOPs for each task. At inference time, they add special instruction and in-context examples to prompt tool using, resulting in extra compute. Because each task contains multiple datasets, we report the average results to represent the general task performance.

## A.2 Models with unknown size

While many of the works use GPT-3.5 or GPT-4 models that do not release their parameter size, we estimate the cost by using the number of tokens processed in extra modules.

**Chameleon (Lu et al., 2023a)**   This work proposes to take a tool-augmented approach to improve on two existing datasets — ScienceQA and TabMWP. Because all experiments use ChatGPT and GPT-4 models, whose parameter sizes are unknown, we only examine results with (the better) GPT-4 model to fairly compare with other methods using GPT-4. Specifically for the ScienceQA dataset, we adopt the Chain-of-Thought (CoT) baseline reported in the paper, and report task accuracy as in the ALL column in Table 3. We calculate the difference in number of tokens between the proposed Chameleon methods against the CoT baseline. For the TabMWP dataset, we adopt the Program-of-Thought (PoT) baseline and similarly calculate the token number difference using the provided results.[3] We adopt numbers in the ALL column in Table 4 as the TabMWP accuracy.

**LATM (Cai et al., 2023)**   This work proposes to use LMs to make tools for individual tasks in BigBench. Compared to the chain-of-thought (CoT) baseline, the proposed LATM method integrates training, validation, and inference stages to make tools and solve questions. We estimate the compute cost by the additional number of tokens used for LATM than for CoT. We measure each method by averaging its accuracy across all six selected tasks.

**CRAFT (Yuan et al., 2023)**   This work uses LMs to make tools for math, table, and image reasoning tasks. We calculate the number of tokens used during training and inference, using its released code and data.[4] CRAFT similarly implements CoT as the baseline, and proposes further training, verification, and finally testing in the CRAFT method. We report its task accuracy on the representative datasets from each task — MATH, TabMWP, and GQA — to enable fairer comparison with other works having overlapping datasets.

**CREATOR (Qian et al., 2023)**   As a prior work for CRAFT, CREATOR similarly tests on MATH and table tasks, but designs its methods differently. In addition to CoT, this work implements a stronger program-oriented baseline called Program-of-Thought (PoT). We also adopt PoT as the main baseline without tool making or using. The CREATOR method operates at test time, with multiple steps through tool making, solution generation, verification, rectification, etc. We calculate the difference in number of tokens between the CREATOR approach and the baseline PoT setting. We adopt the task accuracy reported in Table 2 (MATH) and Table 3 (TabMWP) from the original paper.

**TroVE (Wang et al., 2024b)**   TroVE also induces tools without training supervision. This work adopts the primitive baseline, a presumably stronger version of PoT yet without much textual explanation. The main implementation change in TroVE is the three-mode generation and multi-candidate sampling. We calculate the additional tokens used in TroVE compared to the primitive baseline. The dataset reports task accuracy, solution complexity, and toolbox size, we only adopt the task accuracy to fairly compare with other works.

---

[3]https://github.com/lupantech/chameleon-llm
[4]https://github.com/lifan-yuan/CRAFT

