# OpenReview forum: "What Are Tools Anyway? A Survey from the Language Model Perspective"
_colmweb.org/COLM/2024/Conference — COLM_

### Official Review · Reviewer_XLAS · 2024-04-20

**Rating:** 6
**Confidence:** 4
**Ethics Flag:** 1

**Summary:**

This survey paper provides a unified definition of tools as external programs used by language models (LMs) to enhance their abilities. It systematically reviews various scenarios where LMs use tools, including knowledge access, computation, interaction with the real world, and processing non-textual data. The paper also discusses advanced tool usage methods, tool creation approaches, evaluation benchmarks and metrics, and empirically analyzes the trade-offs between performance gains and computation costs when using different tooling methods across tasks.

**Reasons To Accept:**

- Well-Written and Easy to Understand: The paper is well-organized, clearly written, and easy to follow, even for readers new to the topic of language model tooling. The authors provide clear definitions, illustrative examples, and a logical flow that aids comprehension.

- Comprehensive Coverage: The survey comprehensively covers the landscape of language model tooling, including tools for knowledge access, computation, real-world interaction, and multi-modal processing. It examines basic paradigms, advanced methods, benchmarks, and evaluation practices.

- Insightful Analysis and Future Directions: Through empirical analysis, the paper identifies efficient tooling approaches and tasks that benefit most from tools. It also critically examines missing evaluation aspects like tool reliability and safety, highlighting important future research directions.

**Reasons To Reject:**

My only concern is that, as a survey paper, this work does not present any novel technical contributions or algorithms. Therefore, I am not sure whether it is suitable to publish this paper in COLM?

---

> ### Author Rebuttal · Authors · 2024-05-30
>
> Thank you for your recognition of the clarity, comprehensiveness, and insightfulness of our work!
>
> - **R1: Is survey suitable for COLM**
> As stated in the FAQ of the official COLM website (https://colmweb.org/faq.html): *“We don't have a special track for survey papers. If you wish, you can submit a survey paper and we will route it to the appropriate area chairs and reviewers and based on their judgment, you will get a decision.”*
> From this, we infer that  COLM accepts survey papers and will find the most helpful ACs and reviewers for them. Therefore, we believe our survey paper is appropriate for COLM.

---

> ### Comment · Reviewer_XLAS · 2024-06-04
>
> Thanks for your response, I will keep my score!

---

### Official Review · Reviewer_oLaS · 2024-05-05

**Rating:** 6
**Confidence:** 2
**Ethics Flag:** 1

**Summary:**

This study systematically surveys tools associated with language models (LMs), specifically external programs utilized in LM applications. Beginning with a unified definition of "tools" in the context of LMs, the paper outlines various categories applicable to text-based applications and more sophisticated scenarios. Additionally, the study explores evaluation methods and metrics pertinent to these external programs within the realm of LM.

**Questions To Authors:**

1. Why are the tools in programmatic contexts special and singled out as a section?
2. Can you be more specific on tool creation? Can you give us some guidance on how to create a tool for a specific downstream task based on LMs?
3. What are the benefits of using tools for LMs, compared to other methods like fine-tuning and RAG?

**Reasons To Accept:**

A unified definition of “tools” is proposed for LMs.
A range of relevant works have been listed and categorized in application scenarios.
The evaluation of such “tools” has been discussed, particularly the metrics.

**Reasons To Reject:**

The contribution of this paper may not be sufficient, considering the coverage and the analysis. For example, retrieval-augmented generation (RAG) can also be viewed as an external tool for LMs, particularly, large language models, to generate more accurate content based on given documents.
As for the categorization, it seems to me that “knowledge access”, “Interaction w/ the world”, and “Special-skilled LMs” can be classified as the same functional category, e.g., obtaining external knowledge or information.
In addition, readers may be more interested in identifying bottlenecks for such “tools” and how well they can perform on downstream tasks.

---

> ### Author Rebuttal · Authors · 2024-05-30
>
> Thank you for your recognition of the various aspects covered by our survey!
>
> - **R1: RAG is tool**
> We agree, and **we have explicitly mentioned** that “RAG is a tool” in the paper: we exemplify RAG as a “knowledge access” tool in Table 1, and we discuss RAG in Sec 4.1 the “knowledge access” part.
>
> - **R2: Merge into one category**
> We would like to clarify that **not all these tools are to obtain knowledge**. E.g., make_post() tool from “interact. w/ world” takes actions to change the world instead of obtaining data. So merging these categories would not be appropriate.
> We group tools in a fine-grained way to highlight their uniqueness and provide more insights. E.g., readers can learn that “skilled LM” tools may not always do tasks correctly, which is not true for get_time() from “interact. w/ the world” that always perfectly solves the task. But readers may not grasp this insight if we merge these tools. Nonetheless, readers are free to further interpret based on our discussions.
>
> - **R3: Lack bottleneck discussions**
> We **discuss multiple bottlenecks** of tools across the paper: tasks that tools cannot easily solve in Sec 4.2 and 5.1; tasks that no existing tools can solve in Sec 5.3; limitations in collecting/evaluating tools in Sec 6.1. Lastly, we provide an empirical analysis in Sec 7 on accuracy-efficiency performance, responding to the reviewer's “how well they perform” question.
>
> In general, we believe **our work has sufficient discussions** for the above points regarded as missing by the reviewer, and has clear contributions in definition, taxonomy, evaluation, and empirical study of tools. We would appreciate it if the reviewer could reconsider these aspects. We are happy to take any more feedback!
>
> - **Q1: why programmatic tools**
> Since programming is also a popular medium of tool use, we find it critical to clarify how program and tool relate to each other.
>
> - **Q2: Guidance to create tools**
> Please refer to Sec 5.3 for details on various tool-making testbeds/methods and use the most suitable one. From our empirical analysis in Sec 7, we recommend TroVE when w/ data/compute limitations, and CRAFT otherwise.
>
> - **Q3: Benefits of tools**
> As we stated in Sec1 and Figure1, tools can help LM task-solving (do math accurately) and extend LM abilities (know the current time), which the strongest LM, even with fine-tuning, is fundamentally unable to do. We would like to note that RAG is one form of tool-using, so RAG already benefits from using tools.

---

### Official Review · Reviewer_m17M · 2024-05-10

**Rating:** 8
**Confidence:** 5
**Ethics Flag:** 1

**Summary:**

The authors aim to unify various understanding and definitions of "tools" in connection with language models. This aim is highly relevant and needed to guide interested (lay) people as well as researchers and developers and help them to communicate better.

The paper contains a nice overview on existing definitions and frameworks and evaluations metrics.

**Questions To Authors:**

What exactly does figure 1 show? The caption does not help very much in understanding it. Please provide a better caption. Additoinally, this figure is not referenced in the text. Please do so and provide contextualizing information to the reader: what are they supposed to see in this figure? Figure 4 is also not referenced in the text.

Refering to sections in the paper with the paragraph sign is rather odd. Please use "section 2" etc.

**Reasons To Accept:**

The paper is well written and highly significant. The authors will be able to address the issues raised and submit a revised version.

**Reasons To Reject:**

There are no reasons to reject the paper

---

> ### Author Rebuttal · Authors · 2024-05-30
>
> Thank you for your recognition of the importance and quality of our work!
>
> - **Q1: Better caption for Figure 1**
> Thanks for the feedback. Our figure1 intends to illustrate the text paragraphs around it, to emphasize how (calculator/api) tools come naturally to address two example limitations of current LMs (in math and in getting up-to-date information). We will cite figures1&4 and make their captions clearer in the revised version.
>
> - **Q2: Replace section sign with ‘section’**
> Thank you, we will update this in the revised version!

---

> > ### Comment · Reviewer_m17M · 2024-06-06
> >
> > Thanks for the response!

---

### Official Review · Reviewer_H5ew · 2024-05-11

**Rating:** 6
**Confidence:** 4
**Ethics Flag:** 1

**Summary:**

The paper surveys previous works about LMs with tool-use.
Specifically, it proposes a definition of tool-use as calling methods that are executed externally to the LM.
It then details about types of tool-use and tool types, and advanced tool use scenarios like tool selection and programmatic contexts.
Finally, it presents an empirical investigation of the performance and computation cost of different tool-use approaches.

**Questions To Authors:**

typo on page 3 - SPARL

section 2.1 - "tools Shumaker et al. (2011)" - missing comma

intro - "to obtain time" --> "to obtain the current time"?

abstract - "powerful yet mostly" --> "powerful, yet mostly"?

**Reasons To Accept:**

Tool use is an important subject today when developing language models.
As there is a lot of work on the subject, it may be useful to have such surveys in hand.
The survey covers different aspects of tool use in a useful high-level perspective.

**Reasons To Reject:**

I find the survey and empirical results a bit lacking - I don't think the survey is very comprehensive (e.g. some relevant recent work is missing, for example https://aclanthology.org/2023.findings-emnlp.926/, https://aclanthology.org/2024.eacl-short.10/, https://aclanthology.org/2024.eacl-long.7/). Some earlier works in this area were also missing, e.g. as mentioned in this blog post on the subject: https://newsletter.ruder.io/p/tool-augmented-llms

In general, I don't think I learned enough from the survey - it may be somewhat useful for someone completely new to the research area, but I think it can be improved further by discussing more works and diving a bit deeper into how different tool-use methods work and the pros and cons for each.

---

> ### Author Rebuttal · Authors · 2024-05-30
>
> Thanks for your recognition of the importance and usefulness of our survey paper!
>
> - **R1: survey lacks some works**
> Thank you for pointing out these works!
> First, the three works you pointed out were published within three months of our publication date, so we didn’t get a chance to include them all, but we are happy to add them to our revised version.
> Also, thanks for pointing out the newsletter source. We believe our paper has discussed most of the papers mentioned in it, but we are happy to include the earlier papers left in our revised version. (Specifically, the only tool-related earlier paper left out is *Neural Programmer-Interpreters (Reed and Freitas, 2015)*, and we will add it)
>
> In general, we believe we have provided a comprehensive scope of tool-related topics, and thank you again for providing these additional papers to further enrich the content of our paper!
>
> - **R2: reviewer did not learn enough**
> Thank you for your feedback! Our survey aims to serve as a useful handbook for LM-used tools. Compared to existing methodology or survey works, our paper *uniquely points out* the definition of tools (section 2), a clear taxonomy of tools (section 4), a systematic summary of testbeds (section 5), and most importantly, proposes five essential evaluation aspects (section 6, along with empirical study about them in section 7) that have not yet been evaluated in tool-related literature.
> We particularly discuss “different tool-use methods” in section 3 and 5. We did not spend more content on it due to content length limitations and the importance of other aspects (as mentioned above). However, because tool-use methods to date are mostly homogeneous (e.g., train then test), this lack of diversity may also give the impression of a lack of pros-and-cons discussion.
> In addition to these aspects, if you have other specific topics of interest, feel free to point them out and we are happy to try to incorporate them in our paper.

---

> > ### Comment · Reviewer_H5ew · 2024-06-04
> >
> > Thanks for your clarification. I will keep my score unchanged.

---

### Decision · Program_Chairs · 2024-07-10

**Decision:**

Accept

**Comment:**

The paper presents a survey of tool usage in LLMs, which includes a unifying definition of tools and a categorization of tooling scenarios and methods while considering their computational cost and performance.

Tool usage in LLMs is a timely and growing research area. Surveys that aggregate many existing works from a high-level perspective could be useful in making this research area easily accessible to a wide audience and newcomers.

It seems that researchers already working on tool usage in LLMs may not benefit substantially from this work. While it highlights general challenges and future directions for this research area, these could be made more comprehensive and actionable by considering additional works not covered in the current version of the paper, as well as a more in-depth comparison between existing methods.

[At least one review was discounted during the decision process due to quality]